# Five Predictors Affecting the Prognosis of Patients with Severe Odontogenic Infections

**DOI:** 10.3390/ijerph17238917

**Published:** 2020-11-30

**Authors:** Nathalie Pham Dang, Candice Delbet-Dupas, Aurélien Mulliez, Laurent Devoize, Radhouane Dallel, Isabelle Barthélémy

**Affiliations:** 1CHU Clermont-Ferrand, Inserm, Neuro-Dol, Université Clermont Auvergne, F-63000 Clermont-Ferrand, France; candice.delbet@ch-vichy.fr (C.D.-D.); laurent.devoize@uca.fr (L.D.); radhouane.dallel@uca.fr (R.D.); ibarthelemy@chu-clermontferrand.fr (I.B.); 2CHU Clermont-Ferrand, Délégation Recherche Clinique & Innovation, 63003 Clermont-Ferrand, France; amulliez@chu-clermontferrand.fr

**Keywords:** severe odontogenic infection, cellulitis, allergy, predictor

## Abstract

*Background*: Dental cellulitis management is no longer a simple procedure, as more and more patients are needing long-time hospitalization, several surgeries and intensive care follow-up. This prospective study seeks to highlight criteria that can split patients with severe odontogenic infection into two groups: those with simple evolution and those for whom complex management is necessary. *Methods*: In this observational study, all patients considered with a severe odontogenic infection (which necessitated hospital admission, intravenous antibiotics and general anaesthesia) were enrolled between January 2004 and December 2014 from Clermont-Ferrand University Hospital (France). They were split into two groups: those who needed one surgical intervention with tooth extraction and collection drainage combined with probabilistic antibiotic to treat infection and those who need several surgeries, intensive care unit follow-up or tracheotomy to achieve healing. *Results*: 653 patients were included, of which 611 (94%) had one surgery, 42 (6%) had more than one surgery before healing. Penicillin allergy (*p* < 0.001), psychiatric disorders (*p* = 0.005), oropharyngeal oedema (*p* = 0.008), floor oedema (*p* = 0.004), fever (*p* = 0.04) and trismus (*p* = 0.018) on admission were the most relevant predictors of complex evolution. A conditional inference tree (CTREE) illustrated the association of prognostic factors and the need of multiple surgery. *Conclusions*: Besides clinical symptoms of severity, complications of severe odontogenic infection are predicted by measurables and objectives criteria as penicillin allergy, mandibular molar, C-reactive protein level, psychiatric disorders and alcohol abuse. Their specific association potentialize the risks. IRB number: CE-CIC-GREN-12-08.

## 1. Introduction

A long-time ago, odontogenic infections were considered as a severe or even lethal disease. With the advent of antibiotics like penicillin and improved dental care, they came to be considered as an easily treatable condition. The bases of the therapy are now well known. Typically infections are attributable to decayed or non-vital teeth, postoperative infections, periodontal disease and inflammation of the pericoronal tissues [1,2,3,4]. Surgical incision and drainage of the purulent collections in combination with concerned tooth or teeth extraction, oral cavity rehabilitation and probabilistic antibiotic therapy remain the principles of the treatment [5]. Nevertheless for over ten years, dental cellulitis management is no longer as simple: more and more patients need long duration hospitalizations, several surgical interventions and intensive care follow-up to heal [2,6,7]. The clinical symptoms such as dysphonia, dyspnea, anterior floor edema, limitation of tongue protraction, oropharyngeal edema promptly alert odontologists, oral and maxillofacial surgeons on the severity of the infection but are often misjudged in current practice. Moreover patients with few severe odontogenic infection symptoms may have unfavorable evolution. Many criteria could be involved, the growing number of patients with underlying diseases such as alcohol abuse, immunodepression or long-term diabetes might explain the tendency to more severe infections [2,7,8]. Diagnostic delay with several antibiotics or anti-inflammatory prescriptions may not cure but only lessen the symptoms of an infection. 

The present study was conducted to identify measurables and objectives predictors of adverse evolution during hospitalization for patients with severe odontogenic infection (as several surgical interventions, multiple antibiotic adjustment, intensive care unit follow up and longer hospitalization or death).

## 2. Materials and Methods

The study was designed in compliance with the guidelines of the Declaration of Helsinki, as amended in Edinburgh 2008 and was approved by the Ethical Committee of Rhône Alpes Auvergne CECIC Rhône-Alpes-Auvergne, Grenoble, IRB 5921, IRB number: CE-CIC-GREN-12-08.

### 2.1. Recruitment

Adult patients who meet the following criteria: odontogenic infection which need hospitalization in our department of Oral and Maxillofacial surgery (Clermont-Ferrand University Hospital, France), intravenous antibiotics and surgical management were enrolled in this study between January 2004 and December 2014, but between December 2014 and January 2017 data shall not be useable because of a failure of the software. All patients gave informed written consent. Non-inclusion criteria were pregnancy, limitations of self-expression, and patients under tutorship or curatorship.

### 2.2. Patient Management

Severe odontogenic infections are managed with a standardized protocol. Medical workup was initially performed, including history, clinical examination, C-Reactive Protein (CRP) assay and anesthesiologist consultation. Preoperative images were dental panoramic and cervicofacial computerized tomography-scanner (CT-scan). Airways are secured and patient anesthetized. Aseptic conditions required for any surgery were respected, Povidone betadine scrub 4% (Meda Pharma, Paris, France) was used as antiseptic solution to disinfect the facial, cervical area and oral cavity, Chlorhexidine 2% was used in case of allergy to iodine-based products. Causal tooth was extracted to liberate purulent flow. At this time, sampling of pus was performed. Bacteriological sample are cultured in aerobic and anaerobic conditions. Incision, drainage and debridement were performed for all anatomic cervical fascial spaces involved by infection. Collections were drained with intraoral or transcervical or combined approach. Delbet drains were placed through the opened incisions and retained with a suture to realize large lavages by 0.9% saline. Every day, patients had clinical examination, irrigation and dressing. Postoperative CT-scan, and CRP assay was based on patient’s progress. After surgery, all patients were fed by naso-gastric tube to enable drainage and oral healing.

All patients received intravenous probabilistic antibiotherapy effective against oral mucosa flora. The French Health Products Safety Agency [9] (ANSM) recommendations are monoantibiotherapy with amoxicillin and clavulanic acid (1 g every 8 h); in case of penicillin allergy: clindamycin (300 mg every 8 h) and metronidazole (500 mg every 8 h). Secondly, if necessary, antibiotics were adapted to bacteriological results.

### 2.3. Data Collection

The following data were collected for each patient:(1)Socio-demographic data: age, sex, associate medical condition (penicillin allergy, smoking status, consumption of alcohol, diabetes, asthma, psychiatric disorder);(2)Clinical symptoms of severity;(3)Number and type of space involved by the infection;(4)Treatment on admission: ongoing antibiotic (family and duration of treatment) and/or anti-inflammatory treatment (corticosteroids or Nonsteroidal anti-inflammatory drugs, NSAIDs);(5)Duration of hospital stay and number of surgeries;(6)Bacteriological results, results of culture and sensitivity, antibiotics adaptation;(7)Complications [10].

### 2.4. Statistical Analysis

This study was designed according to results from a retrospective study. Sample size calculation relied on the number of covariates that can be used for adjustment in the multivariable logistic regression model. Based on Harrell et al. [11] work, that suggest a rule of 10 event per variable we required at least 400 subjects with an hypothesised of 10% event rate and an objective to adjust for 4 factors.

Descriptive statistics are expressed as mean and range (min-max), according to their distribution, and number of patients (%) for categorical data. In order to assess the performance of CRP in classifying at-risk patients, namely patients with several surgeries, we used receiver operating characteristic (ROC) curves methods. Area under the ROC curve is plotted with its 95% confidence interval. Besides, sensitivity, specificity, true/false predictive values and positive/negative likelihood ratios are shown at some arbitrary chosen cut-points (>50, >100, >200). Patients at risk to have several surgeries are compared using chi-square test (or Fisher’s exact test when appropriate) for categorical data and using Student *t*-test (or Mann & Withney test when appropriate) for continuous data. A multivariable logistic regression was performed, adjusted for factors that were statistically significant, factors statistically significant but related to clinical symptoms of severity were deliberately not sample in the multivariate analysis because they are often misjudged, only objectives and simple criteria which emerge were taken in consideration. Adjusted odd ratios (aORs) are shown with their 95% confidence interval. A propensity score matching based on the above cited criteria as clinical symptoms gravity, patient characteristics and molar mandibular was performed to identify the complications according to “taking antibiotics” before hospitalization. In other hand, a conditional inference tree (CTREE) was built to illustrate the association of prognostic factors and the need of several surgeries. Statistics were computed with STATA V12 (Stata Corp., College Station, TX, USA) and with R (R version 3.0.2; The R Foundation for Statistical Computing, Vienna, Austria; http://www.r-project.org/). Tests were two-sided and a *p*-value < 0.05 was considered statically significant.

## 3. Results

### 3.1. Baseline Characteristics

A total of 653 patients were recorded, 386 (59%) male and 267 (41%) female with a mean age of 37 years (range 8–88). Of them, 375 (57%) were smokers, 78 (12%) were regular drinkers, 54 (8%) were drug addicts, 62 (9.5%) had psychiatric disorders and 20 (3%) were immunosuppressed. Forty-seven patients (7.2%) were allergic to penicillin. 

The average number of visits before hospitalization in our department was 1.7 (range 0–7). About the course of care, 338 (51%) patients were directly addressed by the dentist, 237 (36%) by the emergency department and 144 (22%) by the general practitioner and 28 (4%) were addressed otherwise. About 189 (29%) patients had consulted two different doctors before being referred to our department (general practitioner, dentist or emergency doctor), 32 (5%) three different doctors and nine patients at least four different doctors (general medical practitioner, general dentist practitioner or emergency practitioner). None of them had had tooth extraction, roots treatment or collection incision.

A total of 378 (64%) patients had received oral antibiotic treatment before presenting to the hospital. The treatment was prescribed by the general practitioner, the dentist or the emergency doctor or due to self-medication. Treatment duration before presenting to the Oral and Maxillofacial Department was 4.6 days (range 1–60). Most patients (231; 61%) were prescribed amoxicillin or amoxicillin and clavulanic acid, 75 (20%) an association of spiramycine-metronidazole, 37 (10%) pristinamycine, 36 (9.5%) metronidazole and 2 (0.5%) clindamycine. Among the 47 patients allergic to penicillin, 33 (70%) patients received antibiotics before admission in our Department but only five (11%) patients had received the first-line antibiotics specifically recommended by ANSM: azithromycine, 500 mg per day or clarithromycine 1000 mg per day or spiramycine 9 MUI per day or clindamycine 1200 mg per day. In the event of treatment failure, metronidazole 1500 mg per day must be add. For all of those patients a new antibiotic therapy was prescribed. A total of 329 (50%) patients had anti-inflammatory treatment orally before presenting to hospital, 50 (7.6%) received corticosteroids, 242 (37%) received NSAIDs and 37 (5.6%) both.

Among 645 patients with collected data, 516 patients (80%) presented during surgery a single facial space involvement by abscess, 95 patients (15%) had two spaces involved, 28 (4%) had three spaces involved, five patients (1%) had four spaces involved and one patient had six spaces involved. The submandibular space and the vestibular space were the most frequently involved, with 32% (*n* = 209) and 28% (*n* = 182), respectively.

Of the 653 patients, 611 (94%) had no further surgery. Forty-two patients (6%) had more than one surgery (range 2–15 surgeries) to drain all collections and their mean hospitalization duration was 14.6 days (range 5–37 days). Among this group, 21 were hospitalized in an intensive care unit, 13 needed a tracheotomy and one patient died of acute respiratory distress associated with sepsis.

Patients who needed more than one surgery to treat the infection represent 6% of the study population, this group includes also all patients with complex evolution (hospitalisation longer than 5 days, hospitalisation in ICU, needed of tracheotomy and death).

### 3.2. Which Are the Risk Factors to Have Complex Evolution?

The results of univariate analysis comparing the 611 patients with single operation and the 42 patients operated several times are displayed in Table 1. The risk factors to have complex evolution are: (1) in the medical history: alcohol abuse, immunodepression, psychiatric disorder and penicillin allergy; (2) during the management: higher rate of consultation, anti-inflammatory and antibiotic prescription without tooth extraction; (3) in the clinical examination: presence of clinical symptoms of severity as trismus, edema, fever, dysphagia, odynophagia, dysphonia, dyspnea, anterior floor edema, orpharyngeal edema, tongue protraction limitation, the tooth and the number of spaces involved by the infection; 4) in the biological results: elevated CRP level. From an arbitrary base CRP level (<50, 50–99, 100–199 and ≥200 mg/L), can estimate the risk to have more than one surgery (Table 2 and Figure 1).

### 3.3. Are Complications Liked to the Antibiotics Taken before Hospitalization?

A total of 378 (64%) patients underwent oral antibiotic treatment before presenting to hospital. Average treatment duration before admission was 4.6 days (range 1–60). Most patients (231; 61%) were prescribed amoxicillin or amoxicillin and clavulanic acid, 75 (20%) spiramycine and metronidazole, 37 (10%) pristinamycine, 36 (9.5%) metronidazole and 2 (0.5%) clindamycine. Among the 47 patients allergic to penicillin, 11 (23%) received no antibiotic therapy and 34 (73%) received antibiotics before admission in our Department; 1/6 (25%) patients with adapted antibiotic therapy, 6/28 (21%) patients with unadapt antibiotic therapy and 1/13 (8%) non-treated patients had unfavourable courses. In the univariate analysis, antibiotic consumption before hospitalisation does not affect the probability to have complications (*p* = 0.39). A propensity score matching based on the above cited criteria as clinical symptoms gravity, patient characteristics and molar mandibular was performed to identify if complex evolution is influenced by the antibiotic treatment prescribed before hospitalization. There is no significant effect of the antibiotics on the same level of severity of infection (*p* = 0.24), even if 50% of patients with favorable evolution versus 35% of patients with unfavorable evolution had had antibiotics before hospitalization. Antibiotic administration before thus hospitalisation does not modify the evolution.

### 3.4. Which Are the Predictors of Multiple Surgeries to Treat Severe Odontogenic Infection?

When focusing on medical history, management and biological results, the multivariate analysis reveals that the risk to have several surgeries in severe odontogenic infection is associated with CRP level: CRP < 50 mg/L (*p* = 0.02, OR = 0.18, IC 95% = [0.05–0.72]), CRP between 50 and 99 mg/L (OR = 1), CRP between 100 and 199 mg/L (*p* = 0.38, OR = 1.55, IC 95% = [0.59–4.13]), CRP > 200 mg/L (*p* = 0.01, OR = 4.12, IC 95% = [1.33–12.72]), alcohol abuse (*p* = 0.03, OR = 2.70, IC 95% = [1.09–6.7]), penicillin allergy (*p* = 0.001, OR = 5.47, IC 95% = [1.99-15-09]), mandibular molar infected (*p* = 0.02, OR = 2.74 IC 95% = [1.16–6.48]). To a lesser extent, it is associated with anti-inflammatory consummation (*p* = 0.06, OR = 20, IC 95% = [1.16–6.48]), psychiatric disorders (*p* = 0.02, OR = 3.02, IC 95% = [1.21–7.55]), immunodepression (*p* = 0.07, OR = 3.32, IC 95% = [0.9–12.31]). Data are reported in Table 3. 

When focusing on clinical symptoms of gravity, the multivariate analysis reveals that the risk to need several surgeries is associated with floor oedema (*p* < 0.001, OR = 5.48, IC 95% = [2.55–11.77]), oropharyngeal oedema (*p* = 0.008, OR = 5.02, IC 95% = [1.52–16.55]), fever (*p* = 0.023, OR = 2.26, IC 95% = [1.12–4.56]), dysphagia (*p* = 0.029, OR = 2.35, IC 95% = [1.09–5.06]) and trismus (*p* = 0.038, OR = 3.24, IC 95% = [1.06–9.87]),

When combining all factors, the most relevant criteria are: penicillin allergy (*p* < 0.001, OR = 6.06, IC 95% = [2.24–16.36]), psychiatric disorders (*p* = 0.005, OR = 3.89, IC 95% = [1.49–10.13]), oropharyngeal oedema (*p* = 0.008, OR = 5.48, IC 95% = [1.54–19.47]), floor oedema (*p* = 0.004, OR = 3.6, IC 95% = [1.51–8.57]), fever (*p* = 0.04, OR = 2.16, IC 95% = [1.02–4.61]) and trismus (*p* = 0.018, OR = 3.99, IC 95% = [1.27–12.51]) on admission.

The study highlighted specific clinical symptoms of severity (such as oropharyngeal oedema, floor oedema, fever and trismus), penicillin allergy, psychiatric disorders and CRP level as predictors of complex evolution. 

### 3.5. Conditional Inference Tree (CTREE) Construction 

In our experience, we have observed that clinical symptoms of severity such as number of spaces involved, dysphagia, dysphonia, dyspnea, tongue protraction limitation, oropharyngeal edema, and anterior flour edema are usually under- or overestimated. In order to exclude those subjective criteria, a sensitivity analysis was performed on two models (one including and one excluding clinical symptoms of severity). The area under the ROC curve is = 0.909 [0.862–0.957] for a more comprehensive model and 0.865 [0.807–0.923] for the simplified model. 

For that reason, CTREE analysis was constructed based the only objective criteria as CRP level, penicillin allergy, immunodepression, mandibular molar infection and psychiatric disorders. It showed that four subgroups of patients with increased risk of multiple surgeries have emerged: (1) patients with a CRP level higher than 200 mg/L have a risk factor of 27% (IC 95% = [14–44], *n* = 37), (2) patients with a CRP level between 50 and 200 mg/L and allergic to penicillin have a risk factor of 30%, (IC 95% = [12–054], n = 20), (3) patients with a CRP level between 50 and 200 mg/L, mandibular molar infection and psychiatric disorder but without penicillin allergy have a risk factor of 33% (IC 95% = [14–44], n = 37), and 4)patients with CRP < 50 mg/L or CRP level unknown and immunodepression, have a risk factor of 25% (IC 95% = [3–65], n = 8). Contrarily, patients with CRP < 50 mg/L or CRP level unknown and any immunodepression have a risk factor of 2%, (IC 95% = [1–4], *n* = 385) to need multiple surgeries. Patients with CRP level between 50 and 200 mg/L without any penicillin allergy presenting a cellulitis not due to a mandibular molar have a risk factor equal to 0 (IC 95% = [0–4], n = 81) to have complications. For patients with CRP level between 50 and 200 mg/L without penicillin allergy nor psychiatric disorders but with infection involving mandibular molar, the risk factor is only 9% (IC 95% = [5–17], *n* = 107) to need several surgical interventions.

Aside from clinical symptoms of severity, CRP > 200 mg/L or CRP = 50–200 mg/L + allergic to penicillin or CRP = 50–200 mg/L + mandibular molar infection + psychiatric disorder or CRP < 50 mg/L + immunodepression increase the risk to have adverse evolution during hospitalization of more than 25% (Figure 2).

## 4. Discussion

The immense majority of patients with severe odontogenic infection have a favorable evolution after a well performed treatment. In our study, only 6% presented a complex course (such as second look, prolonged hospitalization, ICU hospitalization, tracheotomy or death). To gain a better understanding of this group, we looked for predictors of complications based on patient general status, clinical examination, tooth involved and biological markers. The predictors are independent from the pre-hospital management because patients who had or not antibiotic therapy before hospitalization have the same evolution. Moreover, no addressed patients have had tooth extraction and/or collection drainage as reported in the literature [12]. The study highlights specific clinical symptoms of severity as oropharyngeal oedema, floor oedema, fever and trismus are predictor of complex evolution. In the absence of reliable clinical examination, we identified measurable and objective criteria that increase the risk from 25% to 33% to have complications during hospitalization, they are CRP level higher than 200 mg/L, CRP level between 50 and 200 mg/L and allergic to penicillin, CRP level between 50 and 200 mg/L, mandibular molar infection and psychiatric disorder and CRP < 50 mg/L or CRP level unknown and immunodepression.

This study is part of the rare prospective studies about severe odontogenic infection [3,11,12], and with over 653 patients included, it was one of the largest ever reported. Our population with a mean age of 37 years-old, a male dominance, a poor health status, a maximum of mandibular molar infected with submandibular and perimandibular space involvement is similar to the literature [1,3,4,5,6,7,8,11,12,13,14,15,16,17,18,19,20]. Comparing our population to the data reported by the Institut National de Prevention et d’Education pour la Santé (INPES) showed that it is more vulnerable than the general population with a higher proportion of alcohol, nicotine and drug abuse. Reported penicillin allergy was 7% in our study, 4.1% in Zirk et al. study [19] and 8% in the Flynn et al. study [3,6]. In the literature, multiple tooth involvement, multiple space infected, systemic disease, prescription of non-penicillin antibiotics or empirical antibiotic treatment are predictors of hospitalization in patients with acute odontogenic infection [17,21], but few papers report predictors of adverse evolution during the hospitalization.

Clinical symptoms of severity such as number of spaces involved, dysphagia, dysphonia, dyspnea, tongue protraction limitation, oropharyngeal edema and anterior floor edema are well known by experimented oral and maxillofacial surgeons for being pejorative prognostic factors but are sometimes misjudged by other actors. The sensitivity analysis performed on two models (one including and one excluding clinical symptoms of severity) allow us to work on the simplified model. This one has less variable and is based on measurable and objective medical information. CRP level, penicillin allergy, mandibular molar infection, psychiatric disorders and immunodepression allow us to characterize four profiles of patients linked to a risk level higher than 25% to need several surgical interventions and to have complex course.

We started from a principle that CRP is frequently measured on admission when infection is suspected. Its measure is objective and repeatable over time. Daily measurement is useful in the detection of sepsis and it is more sensitive than body temperature or white blood cell count [18]. This inflammatory biomarker rise in most pathological situations associated with inflammation as bacterial, viral infection, trauma, systemic disease flare (excepted lupus) or post-surgical period. Normal human CRP concentrations is less than 1 mg/L. Its level increases in the first 6 h after stimulation by interleukine-6 (IL-6) and can reach peak levels approaching 350–400 mg/L after approximately 48 h; its half-life is of 20 to 24 h [19]. It is commonly considered that mild inflammation and viral infections cause elevation of CRP in the 10–40 mg/L range. Active inflammation and bacterial infection leads to a CRP level from 40 to 200 mg/L. Levels over 200 mg/L are found in severe bacterial infections and burns [20]. Historically, a plasma CRP level of 50 mg/L or more was highly suggestive of sepsis (sensitivity 98.5%, specificity 75%) [18]. We were guided by those considerations to arbitrary define four level of CRP: <50, 50 to 99, 100 to 199 and >200 mg/L. Nevertheless, most studies have failed to establish an objective correlation between CRP level and severity of the sepsis [21]. Meili et al., demonstrated that CRP < 100 mg/L is significantly associated with worst acute respiratory tract infections but have moderate prognostic accuracy in primary care patients to predict clinical outcomes [22]. CRP help to diagnose equivocal cases of appendicitis, cholecystitis, pancreatitis and pelvic inflammatory disease but faster and more interpretable tests are available. Moreover, the cut-off is variable between those different diseases: CRP > 12 mg/L have a sensitivity of 98% to diagnose appendicitis, CRP > 30 mg/L have a sensitivity of 78% to diagnose cholecystitis, CRP levels > 210 mg/L discriminated between patients with clinically mild and severe pancreatitis with a sensitivity of 83% and a specificity of 85% [20]. The CRP level linked with severe infection is variable between the different diseases. The Ylijoki et al., study showed to a statistically significant degree (*p* < 0.001) that CRP concentration on admission is linked with the complicated course of disease, respectively 140.2 mg/L (±67.5) for patients admitted in intensive care unit [1]. However, it is difficult to define an objective CRP level cut-off between patients with severe odontogenic infection with a high risk of unfavorable evolution. It might be necessary to study systematically CRP level in odontogenic infection. Yet, in the inference tree, the level of CRP > 200 mg/L is self-sufficient to have 27% of risk to need multiple surgical intervention with a complex evolution.

In contrast, when the CRP level is lower than 50 mg/L or unknown, immunodepression is self-sufficient to have a risk level of 25% to need multiple surgeries. General health status of patients significantly impacts the course and outcome of severe odontogenic infections. Seppänen et al., showed in their study that among patients with odontogenic infection, 85% of healthy patients developed local complications whereas 75% of medically compromised patients developed systemic infection complications with a need for longer hospital stays and a higher risk of death [8]. In the Optiz et al. study, among 816 patients included, 14 (1.7%) were affected by severe complications after odontogenic infections [7]. All of them had predisposing factors such as diabetes mellitus, obesity, immunodepression and arterial hypertension with its systemic consequences. In this group, long-term alcohol and nicotine abuse where also noticed. In our multivariate analysis, alcohol abuse appeared as a statistically significant risk factor to have severe complications but this factor does not stand out in the inference tree. Tung-Yiu et al., reported among 422 odontogenic infection 11 cases of cervical necrotizing fasciitis, seven of them had immunocompromising conditions [23]. Patients with relevant comorbidities are known to have a worse prognosis and require longer hospitalization compared to patients without concomitant diseases.

If the CRP level is between 50 and 200 mg/L and the patient allergic to penicillin, the risk factor to have multiple surgeries is 30%. Among the 47 patients allergic to penicillin, 33 (70%) received antibiotic therapy but only five received the first-line antibiotics specifically recommended by ANSM before admission in our Department. Still before being admitted, facing the failure of the antibiotic treatment, all patients had a non-justified antibiotic therapy modification and any causal tooth extraction. This contradicts guidelines from the ANSM which state that antimicrobials should be prescribed as an adjunct to removal of the source of infection. Therefore prescription of oral antibiotics without operative intervention represents an inadequate level of treatment for dentoalveolar infections. Prescription of oral antibiotics alone expose patients to more courses of antibiotics which can lead to development of antibiotic resistance. The incidence of routinely antibiotics resistance in deep space head and neck infection is: in the aerobic spectrum, 18% for clindamycin, 14% for macrolides and 7% for penicillin G, in the anaerobes 11% for clindamycin, 6% for metronidazole and 8% for penicillin G [24]. In the Farmahan et al. study resistance to amoxicillin was 26.6% and to both amoxicillin and metronidazole was 18.7% [14], but in a Poeshl et al. study they did not observe any clinical antibiotic failure for patients treated by an association of amoxicillin and clavulanic acid. In contrast, for patients treated by clindamycin, they observed three clinical failures, which necessitated further surgical interventions and a change in the antibiotic regime [24]. The first line antibiotic therapy recommended by ANSM in severe odontogenic infection is a combination of amoxicillin and clavulanic acid. For patients allergic to penicillin, the impossibility to use this first choice is a loss of opportunity. Moreover, the high resistance rate for clindamycin is a matter of concern for those patients. Combination of high resistance rate for clindamycin and of a second choice antibiotic may explain most unfavourable courses of patients allergic to penicillin.

If the CRP level is between 50 and 200 mg/L and the patient not allergic to penicillin, two criteria must be added to obtain a risk factor of 33%: mandibular molar infection and psychiatric disorders. Mandibular molar is the most frequent tooth involved in severe odontogenic infection [3,5,7,11,13,14] and is responsible of infection spreading preferentially in the submandibular space [13,15,17,25,26]. Moreover, in our study it is also strongly correlated with unfavourable evolution. When the point of departure of the infection is the mandibular molar, the submandibular space is often involved, followed by floor oedema and tongue protraction limitation or extension to masseter and pterygoid muscles provoking trismus [25]. Then infection can spread into the parapharyngeal space and be responsible of respiratory distress and dysphagia [25,27] and along jugular and carotid vessels to the mediastinum [10,26,28,29]. The issue then was to understand why infections stay compartmentalised or dispersed. However, mandibular molar infection is not sufficient to have an adverse evolution, and it requires an association with psychiatric disorders. Patients with mental illness have markedly elevated rates of physical disorders compared to the general population. This is widely linked to factors such as lifestyle, adverse psychotropic effects, alcohol and drugs abuses and poor health care [30]. A large review of the literature by Matevosyan et al. reported that patients with psychiatric disorders have more missing teeth, gross caries, decay, periodontal disease and xerostomia [31]. The worst predictive factors of oral health outcomes in this population are old age, type and onset of mental illness. A lack of awareness and a poor perception in oral health are associated with poor oral hygiene and excessive consumption of sugar and lipids. Moreover those patients have poor access to dental care [32,33,34]. The use of treatments or drugs limiting pain perception, late consultation and limited access to appropriate treatment determine the risk for low-noise infection spreading to cervico-facial space and oral health adverse outcomes among adults with psychiatric disorders.

The main limitations of the study concern the absence of evaluation of the precise patients’ comorbid conditions because alcohol, tobacco or drugs consummation is only reported by the patients as penicillin allergy. We considered a group of patients with psychiatric disorders but this group is heterogenous because it concerned patients with minor disorders and institutionalized individuals. Regarding non-steroidal anti-inflammatory drug or antibiotic treatment before hospitalisation, we reported the dose and duration prescribed but the real consumption by the patient was unknown.

## 5. Conclusions

In conclusion, about 6% of patients with severe odontogenic infections may have complications of infections combining multiple surgeries and intensive care unit hospitalization, long-time intubation, tracheotomy and even dying. Measurable and objective presurgical criteria significantly predict the risk to have worse evolution, they allow us to individualize four subgroups at risk of needing multiples surgeries: (1) a group of patients with as CRP level higher than 200 mg/L, (2) a group of patients with as CRP level inferior to 50 mg/L and with immunodrepression condition, (3) a group of patients with an association CRP level between 50 and 200 mg/L and penicillin allergy and (4) a group of patients with an association CRP level between 50 and 200 mg/L and mandibular molar infection and psychiatric disorders. The early identification of patients with a high risk of adverse and complex evolution allow the practitioner to inform his or her patient of risks associated with the disease. It makes possible to anticipate and possibly prevent complications associated with severe odontogenic infection.

## Figures and Tables

**Figure 1 ijerph-17-08917-f001:**
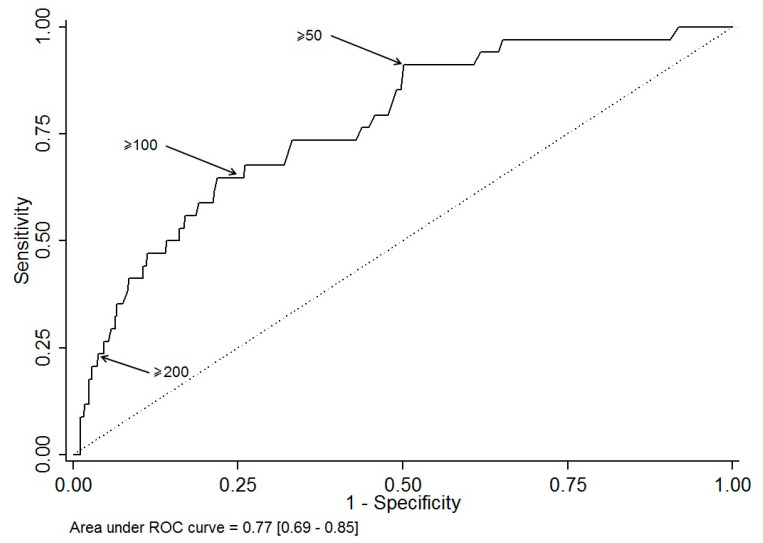
The receiver operating characteristic (ROC) curve for the CRP test results of the study cohort.

**Figure 2 ijerph-17-08917-f002:**
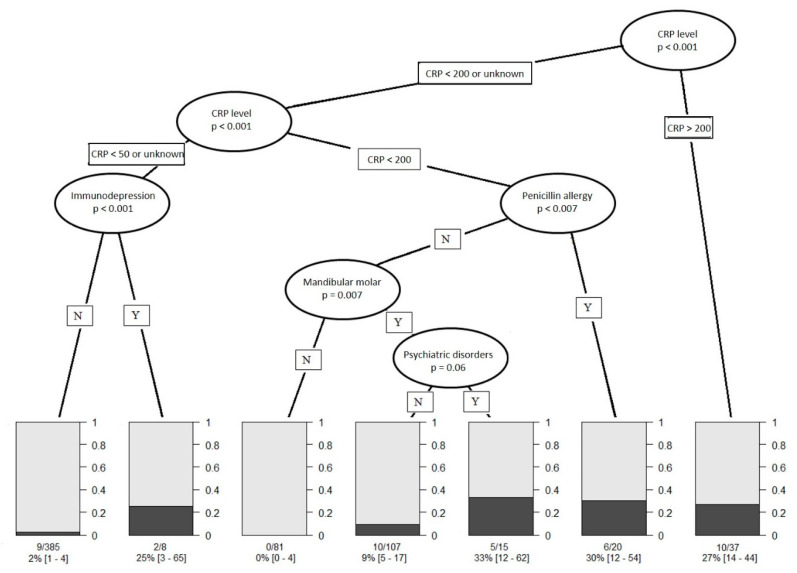
Results of conditional inference tree (CTREE) showing the most predictive indicators to have more than one surgery in case of severe odontogenic infection. Seven nodes where identified (N, no for absent; Y, yes for present). Bar diagrams show the probability to have on surgery in grey and the probability to have more than one surgery in black according the different nodes.

**Table 1 ijerph-17-08917-t001:** Univariate analysis between the two groups: patients which necessitated one surgery (second column) and patients which necessitated more than one surgery (third column) to achieve healing. The fourth column shows the *p* values (* <0.005, i.e., significant effect). Quadrant 1: maxillary incisive and canine, quadrant 2: axillary premolar, quadrant 3: maxillary molar, quadrant 4: mandibular incisive and canine premolar, Quadrant 5: mandibular premolar and quadrant 6: mandibular molar.

Variable	Number of Surgical Interventions = 1 *N* = 611 (%)	Number of Surgical Interventions > 1 *N* = 42 (%)	*p*-Value
Demographic outcomes			
Gender male	360 (59)	26 (62)	0.70
Age	36.8 ± 16.2	39.5 ± 15.6	0.30
Medical history			
Smoker	349 (57)	26 (62)	0.54
Drug abuse	49 (8)	5 (12)	0.38
Alcohol abuse	67 (11)	11 (26)	0.003 *
Immunodepression	15 (2)	5 (12)	0.001 *
Psychiatric disorder	52 (9)	10 (24)	0.001 *
Penicillin allergy	39 (6)	8 (19)	0.002 *
Pre-hospitalization management			
Number of consultations	1.7 +/− 0.9	2.3 +/− 1.5	0.02 *
Anti-inflammatory consummation	297 (54)	32 (78)	0.003 *
Antibiotics consummation	350 (63)	28 (70)	0.39
Clinical symptoms at admission			
Trismus	352 (58)	38 (90)	<0.001 *
No edema	13 (2)	0 (0)	<0.001 *
Facial edema	214 (35)	5 (12)	
Cervicofacial unilateral edema	371 (61)	32 (79)	
Cervical unilateral edema	1 (0)	0 (0)	
Cervical bilateral edema	11 (2)	5 (12)	
Fever	159 (26)	23 (55)	<0.001 *
Dysphagia/odynophagia	127 (21)	27 (64)	<0.001 *
Dysphonia	20 (3)	8 (19)	<0.001 *
Dyspnea	10 (2)	6 (14)	<0.001 *
Anterior floor edema	43 (7)	19 (45)	<0.001 *
Tongue protraction limitation	24 (4)	6 (14)	0.009 *
Oropharyngeal edema	11 (2)	7 (17)	<0.001 *
Biological samples			
C-reactive protein level (mg/L)	72.4 (±71.2)	153.9 (±97.4)	<0.001 *
<50	224 (37)	3 (7)	
50–99	116 (19)	9 (21)	
100–199	86 (14)	12 (29)	
≥200	27 (4)	10 (24)	
Missing values	158 (26)	8 (19)	
**Surgical outcomes**			
Number of spaces involved	1.2 ± 0.5	2.1 ± 0.9	<0.001 *
Tooth position			
Quadrant 1	56 (9)	0 (0)	0.04
Quadrant 2	69 (11)	2 (5)	0.30
Quadrant 3	70 (11)	2 (5)	0.30
Quadrant 4	24 (4)	0 (0)	0.39
Quadrant 5	47 (8)	5 (12)	0.37
Quadrant 6	360 (59)	34 (81)	0.005 *

**Table 2 ijerph-17-08917-t002:** The estimated risk to have more than one surgery according to an arbitrary base CRP level.

Characteristics	Sensibility	Specificity	PPV	VPN	LR+	LR−
CRP level						
≥50	91	49	12	99	1.80	0.18
≥100	65	75	16	97	2.59	0.47
≥200	29	94	27	95	4.93	0.75

Abbreviations: PPV, predictive positive value; PNV, predictive negative value; LR+, likelihood ratio positive; LR−, likelihood ratio negative. A likelihood ratio of greater than 1 indicates the test result is associated with the disease. A likelihood ratio less than 1 indicates that the result is associated with absence of the disease.

**Table 3 ijerph-17-08917-t003:** Multivariate analysis (MVA) between the two groups: patients that necessitated one surgery and patients that necessitated more than one surgery to achieve healing. MVA involves observation and analysis of pertinent data which emerged in the univariate analysis. The second column shows the odds ratio (OR), the third column shows the confidence intervals stated at 95% (CI 95%), the fourth column shows the *p* values (* <0.005, i.e., significant effect). Quadrant 6 corresponds to mandibular molar. C-reactive protein level is dosed in milligrams per liter (mg/L), the interval 50–99 is taken as reference to display the interval CRP < 50 as protective.

Characteristics	OR	CI 95%	*p*-Value
Medical history			
Alcohol abuse	2.70	1.09–6.7	0.03 *
Immunodepression	3.32	0.9–12.31	0.07
Psychiatric disorder	3.02	1.21–7.55	0.02 *
Penicillin allergy	5.47	1.99–15.09	0.001 *
Pre-hospitalization management			
Anti-inflammatory consummation	2.20	0.97–4.99	0.06
Tooth position			
Quadrant 6	2.74	1.16–6.48	0.02 *
Biological samples C-reactive protein level (mg/L)			
<50	0.18	0.05–0.72	0.02 *
50–99	1	-	
100–200	1.55	0.59–4.13	0.38
200	4.12	1.33–12.72	0.01 *
Missing values	0.79	0.27–2.34	0.68

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
