# Peer review of "Five Predictors Affecting the Prognosis of Patients with Severe Odontogenic Infections"

_ijerph, 2020, doi:10.3390/ijerph17238917_

Round 1

Reviewer 1 Report

  1. In abstract - a. methodology part - incomplete. b. the conclusion looks like a result. please modify.
  2. Introduction: gap statement not clear.
  3. Material part started with objective - wrong, please modify.
  4. 2.1 - Data after 2014 to 2020 - long gap. Please provide justification.
  5. Please add, sample size calculation.
  6. Table footnotes are missing.
  7. The limitation part missing.

Author Response

Dear reviewers,

The Authors thank the reviewer 1 for having accepted to read and correct our work.

Please find the corrections and answers we proposed to your comments and suggestions. We hope that it does match with your expectations.

Suggestion 1: In abstract - a. methodology part - incomplete. b. the conclusion looks like a result. please modify.

As recommended, the methodology part is modified as the conclusion. Methodology is complete “In this observational study, all patients considered with a severe odontogenic infection (which necessitated hospital admission, intravenous antibiotics and general anaesthesia) were enrolled between January 2004 and December 2014 from Clermont-Ferrand University Hospital (France). They were split into two groups: those who needed one surgical intervention with tooth extraction and collection drainage combined with probabilistic antibiotic to treat infection and those who need several surgeries, intensive care unit follow-up or tracheotomy to obtain healing” and conclusion changed  “Complications of severe odontogenic infection, is predicted by penicillin allergy, mandibular molar, C-reactive protein > 200 mg/l, psychiatric disorders and alcohol abuse. Their specific association potentialize the risks »

Suggestion 2: Introduction: gap statement not clear.

The objective of the study is more clearly presented: “The present study was conducted to identify predictors of adverse evolution during hospitalization for patients with severe odontogenic infection (as several surgical interventions, multiple antibiotic adjustment, intensive care unit follow up and longer hospitalization or death).

Suggestion 3: Material part started with objective - wrong, please modify.

In the material part, “The aim of this prospective observational study is to identify predictors which may affect the prognosis of patients with severe odontogenic infections” was removed.

Suggestion 4: Data after 2014 to 2020 - long gap. Please provide justification.

Patients were enrolled in this study between January 2004 and January 2017, but between December 2014 and January 2017 data shall not be useable because of a failure of the software.

Suggestion 5: Please add, sample size calculation.

In material and method part, was added “ This study was designed according to results from a retrospective study. Sample size calculation relied on the number of covariates that can be used for adjustment in the multivariable logistic regression model.  Based on Harrell et al. work , that suggest  a rule of 10 event per variable we required at least 400 subjects with an hypothesised of 10% event rate and an objective to adjust for 4 factors.”

Suggestion 6: Table footnotes are missing.

I do not understand this comment because tables have title and footnotes.

Suggestion 7: The limitation part missing.

We added this part :

« The main limitations of this study concern the absence of evaluation of the precise patients’ comorbid conditions because alcohol, tobacco or drugs consummation is only reported by the patients as penicillin allergy. We considered a group of patients with psychiatric disorders but this group is heterogenous because it concerned patients with minor disorders and institutionalized individuals. About non-steroidal anti-inflammatory drugs or antibiotics treatment before hospitalisation, we reported the dose and duration prescribed but the real consummation by the patient is unknown. “

Reviewer 2 Report

  1. Please remove section numbers in the abstract.
  2. Abstract - Methods: Add a brief note on criteria for patient assessment.
  3. Throughout the manuscript, the authors have referenced older studies. It would benefit to introduce the study and discuss it from the perspective of more recent studies reporting predictors for odontogenic infections and its treatment course.

Author Response

The Authors thank the reviewer 2 for having accepted to read and correct our work.

Please find the corrections and answers we proposed to your comments and suggestions. We hope that it does match with your expectations.

To the attention of Reviewer 2:

Suggestion 1: Please remove section numbers in the abstract.

The section numbers are removed

Suggestion 2: Abstract - Methods: Add a brief note on criteria for patient assessment.

We add: “ In this observational study, all patients considered with a severe odontogenic infection (which necessitated hospital admission, intravenous antibiotics and general anaesthesia) were enrolled between January 2004 and December 2014 from Clermont-Ferrand University Hospital (France). They were split into two groups: those who needed one surgical intervention with tooth extraction and collection drainage combined with probabilistic antibiotic to treat infection and those who need several surgeries, intensive care unit follow-up or tracheotomy to obtain healing”

Suggestion 3: Throughout the manuscript, the authors have referenced older studies. It would benefit to introduce the study and discuss it from the perspective of more recent studies reporting predictors for odontogenic infections and its treatment course.

The authors had difficulties to find recent studies.

We added:

Criteria for admission of odontogenic infections at high risk of deep neck space infection

N Alotaibi, L Cloutier, E Khaldoun, E Bois, M Chirat, D Salvan. Eur Ann Otorhinolaryngol Head Neck Dis . 2015 Nov;132(5):261-4.

doi: 10.1016/j.anorl.2015.08.007.

Epidemiological analysis of management of severe odontogenic infections before referral to the emergency department. Konstantinos Katoumas, Dimitrios Anterriotis, Maria Fyrgiola, Violetta Lianou, Dimitrios Triantafylou, Ioannis Dimopoulos. J Craniomaxillofac Surg. 2019 Aug;47(8):1292-1299.

doi: 10.1016/j.jcms.2019.05.002.

Reviewer 3 Report

I would like to thank the authors for their work. In this work, the authors have evaluated five predictors which they believe affect the prognosis of patients with severe odontogenic infections. While the study has a sizable sample size, it is still far from a comprehensive work that can be used as a clinical guideline. As authors have mentioned themselves, they have not included some of the statistically significant factors related to clinical symptoms of severity such as the number of space involved, dysphagia, dysphonia, dyspnea, tongue protraction limitation, oropharyngeal edema, and anterior flour edema in their multivariate analysis. They suggest that these factors are severity risk factors and therefore not included in the multivariate analysis. This can be the case for CRP and the level of CRP can be a severity risk rather than a prognostic factor. I believe that for a comprehensive study, those clinical factors should also be taken into account.

Author Response

The Authors thank the reviewer 3 for having accepted to read and correct our work.

Please find the corrections and answers we proposed to your comments and suggestions. We hope that it does match with your expectations.

To the attention of Reviewer 3,

In our experience, we observed that clinical symptoms of severity such as number of spaces involved, dysphagia, dysphonia, dyspnea, tongue protraction limitation, oropharyngeal edema, and anterior flour edema are well known by the experimented Oral and Maxillofacial surgeons for being pejorative prognostic factor but are sometimes misjudged by other actors. In order to exclude those subjective criteria, a sensitivity analysis was performed on two models (one including and one excluding clinical symptoms of severity). The area under the ROC curve is = 0.909 [0.862 – 0.957] for more comprehensive model and 0.865 [0.807 – 0.923] for the simplified model. The sensitivity analysis performed on those two models (one including and one excluding clinical symptoms of severity) allow us to work on the simplified model. This one has less variable and is based on objective medical information.

Reviewer 4 Report

This paper conducted a prospective clinical study on disease progression and risk factors of severe odontogenic infections. According to the authors, it was one of the largest prospective studies to-date, with 653 patients between January 2004 and December 2014. They identified 4 risk factors that are associated with poor clinical outcomes requiring complex surgical management - CRP > 200 mg/ml; CRP between 50 and 200 mg/l and penicillin allergy; CRP between 50 and 200 mg/l with molar mandibular infection and psychiatric disorders; CRP ≤ 50 mg/l or unknown CRP level and immunodepression.

The study was conducted well. They collected pertinent demographic and clinical data, and analyzed their result with univariate, multivariate, and a conditional inference tree to identify risk factors for severe odontogenic infection. Their analysis was logical, but I think they could have better controlled co-variates, such as number and location of involved teeth and history of previous infection. They identified that both location and # of teeth were significantly correlated with severe disease in the univariate analysis, but they did not consider that these could also be correlated with psychiatric disorders or other co-variates. I think they could have better used their data to try controlling for hypothesized co-variates to see if they can improve their predictions.

  1. Table 1 - Is psychiatric disorder correlated with severe infection if you control for the # and location of infection? It could just be that persons with psychiatric disorders are more likely to have more teeth involved in the odontogenic infection. In general, I suggest adding discussion and analysis on potential confouders or co-variates of the data.
  2. Is the number of teeth involved controlled for in Table 3?

Author Response

The Authors thank the reviewer 4 for having accepted to read and correct our work.

Please find the corrections and answers we proposed to your comments and suggestions. We hope that it does match with your expectations.

To the attention of Reviewer 4,

Suggestion 1: I think they could have better controlled co-variates, such as number and location of involved teeth and history of previous infection.

We reported controlled the number and the location of the involved teeth but we do not collected data about history of previous infection.

Suggestion 2: Table 1 - Is psychiatric disorder correlated with severe infection if you control for the # and location of infection? It could just be that persons with psychiatric disorders are more likely to have more teeth involved in the odontogenic infection. In general, I suggest adding discussion and analysis on potential confouders or co-variates of the data.

We collected the number of untreated caries and dental cyst for each patient, when comparing patients with psychiatrics disorders to patients without psychiatric disorders, the variable “number of untreated caries and dental cyst “ is not statistically different. Because in our population the dental status is not influenced by the psychiatric disorders of the patient, we do not discuss it. But we can change it if you want.

Suggestion 3: Is the number of teeth involved controlled for in Table 3? Yes, it is.

Reviewer 5 Report

Thanks for choose MDPI and IJERPH for publish you manuscript.

Article is well introduced and material and methods are cleary described.

Figure 1 in results is low in quality, modify it or reduce the scale.

To improve the scientifically of your manuscript add in the discussion different techniques or procedures like in ""Clinical Efficacy and Patient Perceptions of Pyogenic Granuloma Excision Using Diode Laser Versus Conventional Surgical Techniques. J Craniofac Surg. 2018 Nov;29(8):2160-2163. doi: 10.1097/SCS.0000000000004734. ""

Normalize ethical and reference to the guideline "Istruction fo the Author"

Author Response

The Authors thank the reviewer 5 for having accepted to read and correct our work.

Please find the corrections and answers we proposed to your comments and suggestions. We hope that it does match with your expectations.

To the attention of Reviewer 5,

Suggestion 1: Figure 1 in results is low in quality, modify it or reduce the scale:

We changed it to increase a little bit its quality.

Suggestion 2: To improve the scientifically of your manuscript add in the discussion different techniques or procedures like in ""Clinical Efficacy and Patient Perceptions of Pyogenic Granuloma Excision Using Diode Laser Versus Conventional Surgical Techniques. J Craniofac Surg. 2018 Nov;29(8):2160-2163. doi: 10.1097/SCS.0000000000004734.

We add those two papers concerning procedure face to dental infection, the first one concerns the criteria for admission of odontogenic infection in hospitalization, it splits severe and not severe odontogenic infection. The second paper reveals that, as in our observation, in most situations, the recommendations for the management of odontogenic infection are not respected.

Criteria for admission of odontogenic infections at high risk of deep neck space infection

N Alotaibi, L Cloutier, E Khaldoun, E Bois, M Chirat, D Salvan. Eur Ann Otorhinolaryngol Head Neck Dis . 2015 Nov;132(5):261-4.

doi: 10.1016/j.anorl.2015.08.007.

Epidemiological analysis of management of severe odontogenic infections before referral to the emergency department. Konstantinos Katoumas, Dimitrios Anterriotis, Maria Fyrgiola, Violetta Lianou, Dimitrios Triantafylou, Ioannis Dimopoulos. J Craniomaxillofac Surg. 2019 Aug;47(8):1292-1299.

doi: 10.1016/j.jcms.2019.05.002.

Suggestion 3: Normalize ethical and reference to the guideline "Instruction for the Author"

We add in material et methods: The study was designed in compliance with the guidelines of the Declaration of Helsinki, as amended in Edinburgh 2008 and was approved by the Ethical Committee of Rhône Alpes Auvergne CECIC Rhône-Alpes-Auvergne, Grenoble, IRB 5921, IRB number: CE-CIC-GREN-12-08.

The references were corrected.

Round 2

Reviewer 4 Report

thank you for addressing my comments.